# Progress in Brain Magnetic Resonance Imaging of Individuals with Prader–Willi Syndrome

**DOI:** 10.3390/jcm12031054

**Published:** 2023-01-29

**Authors:** Zhongxin Huang, Jinhua Cai

**Affiliations:** 1Department of Radiology, Children’s Hospital of Chongqing Medical University, Chongqing 400014, China; 2National Clinical Research Center for Child Health and Disorders, Ministry of Education Key Laboratory of Child Development and Disorders, Chongqing 400014, China; 3Chongqing Key Laboratory of Translational Medical Research in Cognitive Development and Learning and Memory Disorders, Chongqing 400014, China

**Keywords:** Prader–Willi syndrome, morphological MRI, diffusion MRI, functional MRI, brain structure, brain function

## Abstract

Prader–Willi syndrome (PWS), a rare epigenetic disease mapping the imprinted chromosomal domain of 15q11.2-q13.3, manifests a regular neurodevelopmental trajectory in different phases. The current multimodal magnetic resonance imaging (MRI) approach for PWS focues on morphological MRI (mMRI), diffusion MRI (dMRI) and functional MRI (fMRI) to uncover brain alterations. This technique offers another perspective to understand potential neurodevelopmental and neuropathological processes of PWS, in addition to specific molecular gene expression patterns, various clinical manifestations and metabolic phenotypes. Multimodal MRI studies of PWS patients demonstrated common brain changes in the volume of gray matter, the integrity of the fiber tracts and the activation and connectivity of some networks. These findings mainly showed that brain alterations in the frontal reward circuit and limbic system were related to molecular genetics and clinical manifestations (e.g., overwhelming eating, obsessive compulsive behaviors and skin picking). Further exploration using a large sample size and advanced MRI technologies, combined with artificial intelligence algorithms, will be the main research direction to study the structural and functional changes and potential pathogenesis of PWS.

## 1. Introduction

Prader–Willi syndrome (PWS) is a complex neurodevelopmental disease, first discovered by Prader [1] (1956). Based on epidemiological surveys, the incidence of PWS is approximately 1/30,000~1/5000, and the mortality rate is approximately 1.25% to 3% per year [2,3,4,5,6]. PWS is also an epigenetic disorder, mapping to the imprinted chromosomal domain of 15q11.2-q13.3, specifically at the paternal gene expression of the SNORD116 locus [7]. Molecular genetic patterns of PWS mainly include microdeletion (DEL), maternal uniparental disomy (mUPD) and imprinting error [8]. These genetic aberrations cause hypothalamic dysfunction and various endocrine metabolic phenotypes. The clinical manifestations of PWS are initially characterized by infantile hypotonia, poor sucking ability, failure to thrive and hypogonadism. During early childhood, insatiable appetite and uncontrollable obesity occur without appropriate management after approximately 2 years. Other clinical features include obsessive-compulsive (OC) tendencies, behavioral difficulties, mental retardation, neuropsychiatric issues, sleep abnormalities, osteoporosis and scoliosis. The clinical diagnostic criteria for PWS were reported in 1993; molecular diagnostic criteria for PWS were added in 2001; and the latest international diagnostic criteria for PWS were revised in 2011 [9,10,11]. While various clinical and endocrine-related phenotypes contribute to the developmental delay of patients with PWS, brain aberrations are of particular concern. Recently, multimodal MRI studies in PWS have provided useful brain structural and functional information on the underlying pathophysiological mechanisms, allowing for the development of objective and reliable neuroimaging markers. Herein, we reviewed the recent literature on MRI studies of PWS, with the aim of: (i) summarizing the brain MRI methodology of patients with PWS; (ii) understanding the association between brain imaging abnormalities and underlying pathophysiological mechanisms; and (iii) illustrating recent significant findings and limitations to guide future brain MRI research.

## 2. Molecular Genetics

Based on the genetic mechanism, PWS gene expression patterns were classified into four major isoforms: (1) DEL: a microdeletion of the paternal genes at the chromosomal 15q11-q13 locus (approximately 65–70%), further divided into Type Ⅰ and Type ⅠⅠ; (2) mUPD: subsequent to trisomic rescue (approximately 20–30%) in meiosis; (3) imprinting center (IC) mutations in DNA fragments (4%); and (4) balanced translocation of chromosome 15 (<1%) [12,13,14]. IC mutations and balanced translocation are both imprinting errors. Compared to patients with DEL subtype of PWS, patients with other subtypes of PWS are more prone to have psychotic episodes and anxiety symptoms, higher verbal intelligence quotient scores and better social reasoning skills [15,16,17,18]. More importantly, the risk of IC mutations causing psychosis is similar to that in patients with mUPD, and this risk increases to 100% in adults over the age of 28 years [19,20]. For the DEL isoform of PWS, Type Ⅰ is more severe than Type ⅠⅠ, because there are four more genes with deletions [21]. Patients with Type Ⅰ deletion are more prone to skin picking, visual processing deficits and poor academic performance than those with Type ⅠⅠ deletion [15,16].

## 3. Pathophysiological Mechanisms

Regarding the pathophysiology of PWS, there are different theories that mainly focus on two types. The most common theory is gene-induced disruption of the hypothalamic pathway. In patients with PWS, infundibular nucleus (INF) and paraventricular nucleus (PVN) alterations were observed in postmortem hypothalamic tissue specimens [22]. The two components of hypothalamus have both orexigenic and anorexigenic neuronal populations, participating in the regulation of feeding behaviors [23,24,25]. In addition, hypothalamic-pituitary axis-related peptide hormone-expressing neurons are involved in energy metabolism [23,24,26,27]—that is, dysfunction of hypothalamic neuropeptides and hormones results in the metabolic phenotype of PWS, including extreme overeating, obesity and other clinical features [28,29,30,31]. Contrary to the above study results, another study found no abnormalities in related neurons of the hypothalamus in patients with PWS [32]. Of note, another theory about excitation-inhibition imbalance mechanism focuses on classical transmitters (i.e., amino acids, acetylcholine and monoamines). Classical transmitters are not closely linked to the hypothalamus and are used to understand clinical phenotypes besides obesity, such as OC behaviors, psychiatric problems and some autonomic deficits, in patients with PWS [33,34]. Gamma-aminobutyric acid (GABA) is an amino acid. Genes encoding of GABA type-A (GABA_A_) receptor are localized on chromosome 15 [35]. Therefore, GABA_A_ receptor composition and expression in patients with PWS can affect neurobehavioral function, associated with impaired responses to somatic pain, and result in a high propensity for psychiatric symptoms [36,37]. Further measurements of GABA levels in the brain are needed to prove this hypothesis. Monoamines, such as those in 5-hydroxytryptamine (5-HT)-, noradrenaline- and dopamine-containing pathways, may be associated with reward and addiction circuits involved in behavioral problems, such as skin picking.

## 4. Clinical Manifestations

The primary clinical features of PWS are developmental abnormalities, including behavioral difficulties, cognitive and neurologic aberrations, intellectual disabilities, some atypical manifestations and comorbidities [38]. Classical nutritional stages in PWS are poor feeding in early neonatal life and consequent overwhelming appetite leading to morbid obesity [39,40,41] (Table 1). Severe hypotonia may cause decreased fetal movement and failure to thrive during infancy. During childhood, patients with PWS show a decrease in muscle mass and an increase in fat mass, even in nonobese persons [42,43], a phenomenon that tends toward centripetal obesity in adulthood. Hyperphagia in PWS manifests as an extreme desire to eat, a lack of normal satiety and confusion about food choices [44,45,46]. A prior study suggested that excessive preoccupation with food might be an overexpression of the joviality of eating [47]. Another study suggested that abnormal eating behavior in patients with PWS may be due to reduced satiety, rather than increased hunger [48]. Distinctive facial features in patients with PWS are also found, with a narrow bifrontal diameter, almond-shaped palpebral fissures, strabismus, a thin upper lip with a downturned mouth and enamel dysplasia [39]. Children with PWS may take twice as long to achieve physical and social milestones compared to healthy individuals, then progress to having mild intellectual disability [49]. PWS also results in sleep apnea, reduced pain sensitivity, decreased gastrointestinal motility and scoliosis. In addition to primary developmental disorders, other psychiatric conceptions of mental and behavioral phenotypes include temper outbursts and anxiety, OC behaviors, rigidity and social cognition deficits [19]. Obsessive compulsive (OC) behaviors in patients with PWS, such as skin picking, bring pleasure and comfort, which are slightly different from the typical tendencies of OC disorder [50]. These mental health conditions in patients with PWS overlap with each other, and the overlapping neurobiological mechanisms and genetic factors are difficult to isolate.

## 5. MRI

Recently, with the development of neuroimaging methods, multimodal MRI has been mainly used to study the structural and functional brain changes in binge eating and OC behavior in patients with PWS, revealing the underlying pathophysiological mechanisms from a macroscopic perspective. These findings provide an important role for further treatment, care and prognosis. Hence, the PWS-related brain regions and the PWS-related MRI methodology are both important for understanding the pathophysiological mechanisms of PWS-related phenotypes.

### 5.1. PWS-Related Brain Regions

PWS-related brain regions are generally delineated by neuroimaging-based atlas. Neuroimaging studies of PWS have focused on three main regions of the brain: (1) the limbic dopamine system, which is involved in reward learning and affected-driven motivation [(i.e., the orbitofrontal cortex (OFC), striatum, hippocampus/parahippocampal gyrus, amygdala (AMY) and anterior insula)]; (2) cortical cognitive control circuits that are involved in emotion control and the inhibitory regulation [(i.e., the dorsolateral and medial prefrontal cortex (DLPFC and MPFC), anterior cingulate cortex (ACC) and insula)]; and (3) the hypothalamus, which is involved in energy homeostasis balance and neuroendocrine alterations (explained in detail in the Section 3).

The frontal and limbic brain regions are associated with hunger, satiation and reward processing. The prefrontal cortex (PFC), including the dorsal PFC, ventromedial PFC and ventrolateral PFC, is involved in the regulation of limbic reward and higher-order cognitive functions (i.e., attention, inhibitory control, decision-making, emotion and motivation). The dorsal PFC maintains up-down processing and extends to meta-cognitive control over food intake decision-making. The dorsal PFC plays a vital role in executive functions, memory formation and flexibility toward overeating and OC behaviors in patients with PWS [51]. The ventromedial PFC is related to the regulation of negative emotion; responses to salience attribution and stimuli, such as motivational processing; self-referent disinhibition; and visceral signaling [51]. The ventrolateral PFC is involved in automatic response impulsivity [51]. Additionally, the connections among frontal-subcortical regions are an important part of the reward process. For instance, the OFC and AMY both participate in eating motivation [52]. The OFC is closely connected to the MPFC, responding to food-related sensory information, visceral perception and information integration, as well as engaging in reward learning [53]. The AMY is a subcortical structure and plays a vital role in the arousal of the internal state, especially the arousal of both food hunger and pleasure [54]. More broadly, ACC damage results in imbalances in emotional and cognitive processing, as well as in an increased risk of overeating [55]. The insula cortex governs the primary gustatory sensation, regulates the endosensory system and is responsible for the complex clinical manifestations of PWS, such as pain, itching, mood and hunger [56].

### 5.2. PWS-Related MRI Methods

Neuroimaging methods have been used to reveal brain abnormalities in PWS, including positron emission tomography (PET), magnetic resonance imaging (MRI) and magnetoencephalography. Among them, brain MRI in patients with PWS has the advantages of being radiation-free, providing multiparameter options, and having high tempo-spatial resolution. The MRI methodological flow chart of PWS is shown in Figure 1.

#### 5.2.1. Conventional Structural MRI

Initially, MRI studies of PWS were primarily based on anatomy and morphology, revealing general cortical atrophy and gyrification, corpus callosum agenesis, cerebellar abnormalities, light ventriculomegaly and pituitary hypoplasia [57]. Diffuse brain atrophy is accompanied by ventriculomegaly, including white matter (WM), gray matter (GM) and the total cortical surface. Sylvian fissure abnormalities occur with incomplete insula closure, which is associated with language impairment [58]. Corpus callosum (CC) agenesis is commonly seen in hereditary intellectual retardation, and cerebellar abnormalities may be associated with hypotonia [59]. Pituitary hypoplasia is defined as decreased pituitary height and the absence of or a reduction in bright spots in the posterior pituitary and empty sella [60,61]. However, these results are not specific for PWS and vary among studies, mainly due to insufficient samples, inconsistent maturation of brain regions and different subtypes and imaging processing technologies. Additionally, few studies have directly linked these neuroimaging findings to specific manifestations of cognition and development. Therefore, further large sample, quantitative and qualitative MRI studies are needed to locate the brain regions and brain networks and their clinical correlations in PWS.

#### 5.2.2. Morphological MRI (mMRI)

The majority of quantitative structural MRI studies in the brains of patients with PWS have used voxel-based morphometry (VBM) and surface-based morphometry (SBM) to evaluate microstructural damage by measuring various metrics, such as GM volume, cortical thickness and cortical folding patterns. A number of researchers in PWS studies have highlighted dysfunctions in cortical and subcortical structures involved in satiety and reward (Table 2).

Reduced volume in brain regions may disrupt normal functions of neural pathways and neural networks. Ogura et al. [62] used VBM to find obvious decreases in the volume of the OFC and caudate nucleus, with or without total GM volume as a covariate. Caudate nucleus damage exhibited neurobehavioral abnormalities similar to frontal pathology, suggesting that the dysfunction of the orbitofrontal-subcortical neural network plays a crucial role in overeating and OC behaviors in PWS [70]. Another PET study found enhanced metabolism in the OFC and striatum in obese samples, which also supports altered brain OFC-subcortical neural pathways [71]. Xu et al. [55] concentrated on uncontrolled food intake and cognitive-emotional dysregulation and found cortical volume reduction in 10 covarying brain regions, in which the DLPFC/MPFC and ACC comprise the default mode network (DMN). Moreover, decreased hypothalamus volume was observed in this study. Hypothalamic damage leading to impaired satiety control, disruption of reward circuits and dysregulation of inhibitory control could be responsible for extreme adiposity and obesity in patients with PWS [72].

Contrary to the above findings, other studies have found an increase in the volume of some brain regions. Manning et al. [66] used VBM to explore the brain structure in youth with PWS and first observed widespread increased GM volume, a phenomenon that is largely due to increased cortical thickness. Caixàs et al. [69] also reported increased GM volume in sensorimotor and subcortical regions, which could result in imitation impairment of motor actions in adults with PWS. Another structural MRI study found similar increased cortical thickness, thus confirming delayed cortical maturation in children with PWS of the mUPD subtype [64]. Cortical thickness alterations might indicate neuronal migration and impair elimination of ineffective synapses (pruning). In addition, the local gyrification index (lGI) was used to quantify the abnormalities of the cortical complex, which is related to cortical surface area rather than cortical thickness. Reduction in lGI in multiple brain regions might reflect the developmental disturbance in intracortical organization and cortico-cortical connectivity in developmental delay or mental delay [73]. Lukoshe et al. [65] found that children with PWS have widespread low lGIs in the frontal, temporal and parietal lobes bilaterally. These widespread alterations in bilateral cortical complexity also mean that PWS has a pathological basis for cognitive impairment and developmental delay.

Cerebellar volume alterations are also one of the focuses of many PWS studies. PWS patients have a relative decrease in posterior inferior lobule volume and a relative increase in deep cerebellar dentate nuclei volume, reflecting bulimia, OC behavior, autism spectrum disorder (ASD) and mental retardation [68]. Cerebellar hypoplasia results in motor syndrome and cerebellar cognitive affective syndrome (CCAS), which is consistent with previous gross autopsy results [62,74]. CCAS is mainly associated with damage to the posterior and middle lobules of the cerebellum and has important effects on spatial cognition, executive function, language processing and endostatic dysregulation [75].

Additionally, different genetic subtypes at different age periods have been found, although with some differences. Honea et al. [63] performed the first neuroimaging study on the genetic subtype of adults with PWS and found extensive GM and WM volume decreases in these patients. This study found that the GM volume was more extensively reduced in patients with the DEL subtype than in those with the mUPD subtype, suggesting that patients with the DEL subtype may maintain normal neuronal activity by altering the synaptic compensation mechanism of GABAA receptor binding. In addition, the patients with the mUPD subtype in this study were younger than those with the DEL subtype, which can confound the results. Subsequently, a further subtype analysis of PWS controlled for subjects aged from 6 to 18 years and emphasized the separated neurodevelopmental patterns [73]. This study showed that patients with the mUPD subtype had more severe early brain atrophy, thickened cortical thickness and DMN changes than those with the DEL subtype, suggesting a greater risk of psychiatric manifestations [64]. In contrast, the DEL subtype exhibits a proportional reduction in GM volume, without signs of cortical atrophy but with fundamentally arrested development [64]. Ge et al. [67] concentrated more on the neonatal population with PWS and thought that changes in the frontal areas (i.e., the middle frontal gyrus, OFC and inferior frontal gyrus) result in differences in maladaptive behaviors and emotions between the two subtypes.

Notably, brain development is a complex process, involving neuronal proliferation and migration, establishment of synaptic connections, myelin formation and elimination of ineffective synapses (pruning) [76]. Metrics, such as cortical surface area, cortical thickness, GM volume and cortical folding patterns, correspond to overlapping pathological mechanisms of developmental abnormalities. Different genetic subtypes of PWS may have different patterns of brain development, with patients with the DEL subtype having poorer learning ability in childhood and patients with the mUPD subtype being more prone to psychiatric symptoms in adulthood. Different time points and brain area measurements of PWS samples tend to confound morphological findings. Therefore, decreased basal ganglia (i.e., the caudate nucleus and the lentiform nucleus) volume and increased ACC volume are in contrast to the findings of previous studies [62,63].

In conclusion, mMRI-based studies of PWS are gradually expanding from the cerebral cortex to subcortical GM, from the supratentorial brain to the subcortical cerebellum, from simple volume to cortical thickness, and even including cortical gyrification. Although there have been some achievements in the study of PWS morphology, the results are quite different, mainly due to the differences in sample size, age range, imaging techniques and analysis methods. Moreover, most of these results are based on the analysis of differences between groups; these results cannot be used for the personalized diagnosis of PWS. Machine learning is a promising avenue through which the problem of individualized diagnosis of PWS, using imaging data, can be solved by establishing a classification model.

#### 5.2.3. Diffusion MRI (dMRI)

Diffusion tensor imaging (DTI), a major part of dMRI, can be used to quantitatively assess the integrity and orientation of WM fiber tracts in vivo and is currently one of the most effective methods for studying structural connectivity in WM. Fractional anisotropy (FA) is the most common index for evaluating microstructural damage in WM fiber bundles. Existing studies have focused on changes in DTI diffusion parameters reflecting WM fiber damage that may be developmentally, behaviorally and genetically related, providing a basis for diagnosis and progression assessment of PWS. Further details of dMRI articles about PWS are summarized in Table 3. Combining mMRI and dMRI, we can confirm the structural brain alterations in patients with PWS, from GM and WM, to better understand the cognitive and developmental abnormalities of the disease from multiple perspectives.

Yamada et al. [77] first observed objective evidence that patients with PWS indeed exhibit a unique maturational delay in axonal structure and provided a new prospective for understanding the pathophysiology of PWS. This study revealed decreased FA values in the bilateral posterior limb of the internal capsule (PLIC), corpus callosum (CC) and right frontal WM and increased trace values in the left frontal WM and left dorsomedial thalamus in PWS patients [77]. These diffusivity characteristics correspond to the clinical manifestations. In particular, bilateral PLCL abnormalities hint at motor neuron lesions, which is highly consistent with central hypotonia in PWS [80]. After that, Rice et al. [78] performed a whole-brain voxel-wise analysis of 15 PWS patients and 15 healthy controls based on tract-based spatial statistics (TBSS). Among the results, WM alterations in the splenium of the CC and the left inferior frontal occipital fasciculus (IFOF) were not unique to PWS and can also be found in patients with ASD, attention deficit hyperactivity disorder (ADHD) and other genetic abnormalities. Splenium damage is responsible for the deficit in attentional transitions and the aversion to change. In addition, posterior thalamic radiation plays a key role in sensorimotor disorders (e.g., hypotonia and reduced pain sensitivity), and the IFOF plays a similarly important role in semantic processing and emotion recognition. In the same year, Lukoshe et al. [79] also used TBSS to concentrate more on the WM microstructure of different subtypes of PWS at similar ages. New findings revealed damage to the WM fiber bundles of the cingulum, superior longitudinal fasciculus, anterior and superior corona radiata, and external capsule in patients with the mUPD subtype. Aberrant WM microstructure in patients with the mUPD subtype overlaps highly with psychoses, such as schizophrenia, and correlates weakly with developmental delay, suggesting that delayed myelination might result from primary abnormalities in neurons rather than primary oligodendrocyte dysfunction. Xu et al. [55] used probabilistic tractography analysis based on 10 covarying brain regions to demonstrate both WM and GM lesions, explaining that WM integrity degradation and GM damage are related to overeating and constant hunger in patients with PWS.

At present, there are relatively few DTI results of PWS, and the consistency is poor, mainly because of the small sample size, the lack of age grouping and cross-sectional studies. Future work focusing on large samples, and longitudinal studies of the relationship between WM microstructure and clinical phenotype in patients with PWS, is essential.

#### 5.2.4. Functional MRI (fMRI)

The most widely used noninvasive functional neuroimaging technique is blood oxygen level-dependent fMRI (BOLD-fMRI). BOLD-fMRI uses hemodynamics to express neural metabolism, meaning that the coactivated BOLD signal is used to represent synaptic currents and action potential propagation. The technique consists of two main components: task fMRI (t-fMRI) and resting-state fMRI (rs-fMRI). t-fMRI was used to observe the activation of brain regions by applying task stimuli to subjects. rs-fMRI measures the brain blood oxygen level of individuals in a resting state, reflecting the underlying state of brain activity, in which the brain regions with the same activity are considered to constitute a coordinated functional brain network. Both t-fMRI and rs-fMRI have been applied in PWS research (Table 4 and Table 5).

##### Overeating

Initially, t-fMRI studies on PWS focused on food-related homeostatic and visceral networks with food stimuli. Miller et al. [83] found enhanced activation in ventromedial prefrontal cortex (VMPFC) after ingestion of oral glucose load in patients with PWS. Holsen et al. [82] found widespread altered food motivation networks based on visual food stimuli in the PWS group during pre-meal and post-meal states. These brain regions are the AMY, OFC, MPFC, insula, parahippocampal gyrus and fusiform gyrus. The post-meal-activated brain regions (i.e., OFC, hippocampus, parahippocampus) were similar to the results of another mMRI study [63]. It was confirmed that the failure of the syndrome mechanism associated with the OFC and the alteration of the influence-driven motivation associated with the limbic system are obvious in patients with PWS. Holsen et al. [85] also observed functional differences in divergent PWS genetic subtypes before and after visual food stimuli, with DEL showing hyperactivations of food motivation networks (MPFC and AMY) and mUPD showing hyperactivation of inhibitory control networks (DLPFC and parahippocampal gyrus). Using the same approach, the authors further studied the simple obesity and PWS populations, with hyperactive subcortical reward circuitry and hypoactive prefrontal inhibitory regions in patients with PWS compared to individuals with simple obesity (especially post-meal), supporting PWS as a model for extreme obesity [86]. Similar results were obtained by Shapira et al. [81] based on temporal clustering analysis (TCA), which showed that glucose ingestion results in a delay in the hypothalamus and other satiety networks, such as the nucleus accumbens, insula and MPFC. Dimitropoulos and Schultz [84] studied nine PWS patients and 10 controls under visual food stimulation during the hungry state, using general linear model (GLM) analysis, and concluded that the hypothalamus and OFC are overactivated in response to high-calorie food stimulation, consistent with PWS patients’ fondness of sweet food [92]; however, this is in contrast to previous studies, using positron emission computed tomography (PET), which showed patients were insensitive to high-energy foods [93,94].

To investigate the underlying mechanisms of PWS more accurately, recent studies have preferred to use rs-fMRI without consideration of food stimuli to study the brain functional connectivity and brain networks of excessive eating behavior. Zhang et al. [88] selected brain regions with reduced amplitude of low frequency fluctuations (ALFF) values as ROIs for four neural networks (the DMN, core areas, prefrontal areas and motor sensory areas) and found altered functional connectivity among these brain regions, providing further evidence for a model of overeating, and even addiction, in patients with PWS. Zhang et al. [89] then combined ALFF and Granger causality analysis (GCA) to study the brain network in patients with PWS, and the results hint at enhanced causal influences from the AMY to the hypothalamus and from both the MPFC and ACC to the AMY, leading to extreme overeating in patients with PWS [89]. This provided another neural perspective to better understand the causal influence between excessive food motivation and hyperphagia consistently in children with PWS.

##### Other Behaviors

Klabunde et al. [87] recruited 10 individuals with PWS and recorded skin picking and non-picking episodes and found that skin picking severity was significantly negatively correlated with the activation of the right insula and left precentral gyrus. Both the high pain threshold and frequent itching-related regions in patients with PWS are nearly identical, concentrating on interoceptive, motor, attention, somatosensory and reward processing regions [87]. Previous studies in patients with PWS also found that increased functional connectivity in the anterior cingulate/insula, and GABAA receptors with high-affinity benzodiazepine binding sites, were significantly reduced in the limbic system [27,88]. Thus, OC phenomena in patients with PWS, such as skin picking and compulsive eating, appear to be linked to the insula and sensorimotor cortex [81,90]. Pujol et al. [90] observed an enhanced primary sensorimotor cortex-basal ganglia loop and dysfunctional frontal-basal ganglia circuits in patients with PWS, based on the caudate nucleus and putamen nucleus as seed points. That is, the disinhibition or facilitation of the sensorimotor cortex is more direct than the prefrontal cortex to the stimulus-response pathway, clearly explaining compulsive eating in patients with PWS. Tauber et al. [91] found increased connectivity of the right superior orbitofrontal network after oxytocin treatment in infants with PWS, based on independent component analysis (ICA), which was correlated with improved oral motor skills and social behaviors. This was the only MRI study that focused on treatment.

Functional dysconnectivity in many brain regions in patients with PWS has been demonstrated, and whether the connectivity deficits in these regions are independent requires more network analysis based on graph theory. Previous studies have mainly focused on networks constructed based on a priori ROI, but the relationships within and between networks are intricate and complex. A purely data-driven approach has been used to restore the relationship between real hemodynamic responses and neuronal synchronization and to identify driving factors of neuropathy and abnormal energy metabolism in patients with PWS. In addition, PWS still has small-world attributes, but the whole-brain networks change in terms of the nodal and global properties representing its graph-theoretic metrics, which should be the focus of future research.

## 6. Challenges and Future Prospects

Neuroimaging by means of MRI can intuitively and noninvasively help detect brain structural and functional abnormalities and explain the genetic and pathophysiological mechanisms in PWS. Altered brain regions and brain connectivity might be used as biomarkers to distinguish cognitive and developmental abnormalities caused by PWS or other factors. At present, the challenges of PWS studies include the following three main components. First, due to the different clinical symptoms and inconsistent concerns of patients with PWS at different ages, the recruited PWS sample had milder behavioral abnormalities in children and more severe symptoms in adults, and the existing cognitive and behavioral scales could not comprehensively and objectively reflect the actual cognitive profile of PWS in all age groups. Behavior and cognitive assessment scales for children with PWS in the primary age group remain unclear and involve many difficulties. Second, PWS is a rare disease with low disease prevalence, and the inclusion of subjects has the problems of a large age range and small sample size; even after balancing age and sex, confounding factors still reduce the accuracy of measurements. Therefore, for a small sample size and large age range in the PWS population, the pairing principle should be strictly followed to control errors. Third, recent research in PWS has focused more on intake and OC behavior and less on longitudinal studies in younger age groups, pre- and post-treatment comparisons of growth hormone, and other cognitive impairment and comorbidity studies. Among them, brain alterations before and after PWS drug treatment should be focused on, to facilitate future prognosis and management. Studies on the psychiatric mechanisms of PWS have gradually intensified in recent years [95], and this is an issue that needs to be explored in future research.

Recently, advanced MRI technologies combined with artificial intelligence (AI) have progressed rapidly. In addition to the previously described mMRI, DTI and BOLD-fMRI, other functional and structural imaging techniques may also reveal the underlying pathophysiological mechanisms of PWS. In terms of sMRI sequences, data obtained with advanced dMRI, such as diffusion kurtosis imaging (DKI) and diffusion spectrum imaging (DSI), might better explain how white fiber bundles provide more abundant information about the non-Gaussian distribution of water molecules. For fMRI sequences, arterial spin labeling (ASL), MR spectroscopy (MRS) and susceptibility-weighted imaging (SWI) can monitor perfusion, cellular metabolism and iron-containing heme deposition, which may provide objective and reliable neuroimaging biomarkers for the clinical evaluation of PWS. Structural and functional MRI showed that extensive changes might occur in PWS brain circuits and related networks, and further studies of the disruption of brain network integration based on graph theory are necessary. In addition, algorithms for machine learning and deep learning-assisted diagnosis of PWS can improve both the accuracy of diagnosis and the precision of treatment. Although the application of AI in the medical field is still in its infancy, the trend is to diagnose and manage PWS populations with the help of big data.

## 7. Conclusions

In light of the recent research evidence, structural and functional MRI methods can reflect brain changes in patients with PWS in language, motor and memory areas, not only in the cerebral cortex and subcortical gray matter nuclei, but also in the WM fibers surrounding them. Although the causes of behavioral and developmental abnormalities in PWS remain unclear, the prefrontal reward circuit and the limbic system are important regions influencing cognitive function. In the future, the brain alterations in patients with PWS are expected to be investigated in studies in which the sample size is larger and multimodal MRI technologies are applied and combined with AI algorithms, to further explore the effects of developmental and behavioral alterations on cognition, emotion, language and movement. In conclusion, we believe that PWS causes functional and structural brain damage that may be related to underlying diffuse neuroendocrine and neurochemical systems, which will eventually be clarified and will provide guidance for the early evaluation and intervention of PWS.

## Figures and Tables

**Figure 1 jcm-12-01054-f001:**
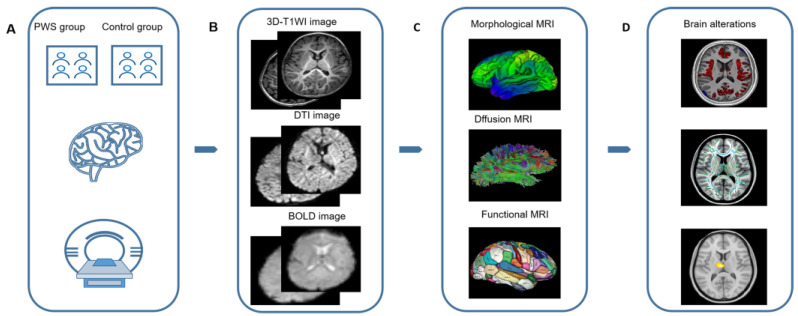
Schematic figures of brain magnetic resonance imaging (MRI) analysis of individuals with Prader–Willi syndrome (PWS). (**A**) The PWS and control groups are included and MRI scans are performed. (**B**) The raw MRI acquisition images are obtained. (**C**) Data processing are performed on the acquired MR images. (**D**) Statistical differences in brain structure and function are obtained by comparison between groups.

**Table 1 jcm-12-01054-t001:** Clinical characteristics of the nutritional phases observed in patients with PWS.

Phase	Period	Clinical Characteristics
Phase 0	Before birth	Decreased fetal movements and low birth weight
Phase 1a	0–9 months	Feeding difficulties (poor sucking and hypotonia) requiring gastric or nasogastric tubes Decreased appetite
Phase 1b	9–24 months	Feeding difficulties have lessened and weight might increase at a normal rate
Phase 2a	2–4.5 years old	Begin to gain weight without changes in appetite or feeding Restriction to 60–80% of recommended daily allowance for calories is needed to prevent obesity
Phase 2b	4.5–8 years old	Appetite and weight acceleration Global developmental delay
Phase 3	Starting from 8 years old	Lack of satiety is obvious and classic gluttony also becomes prominent Onset of mild intellectual disability and food-related temper tantrums
Phase 4	After being an adult	Only some adult patients with PWS control food seeking behaviors and temper tantrums

**Table 2 jcm-12-01054-t002:** Studies of mMRI in PWS.

Number	Author	Country and Equipment	MRI Assessment Method	Subjects (*n*) Total/PWS	PWS Mean Age (Years)	Key Findings
1	Ogura et al. [62]	Japan GE 1.5 T	VBM	25/12	23.7	- PWS vs. HC: showed small GM volume in the OFC, caudate nucleus, inferior temporal gyrus, precentral gyrus, supplementary motor area, postcentral gyrus and cerebellum. - PWS vs. HC: showed small GM volume in the OFC with total GM volume as a covariate.
2	Honea et al. [63]	USA Siemens 3.0 T	VBM/ GLM	48/23 5 DEL I 10 DEL II 8 mUPD	25.2 (DEL) 17.4 (mUPD)	- PWS vs. HC: showed lower GM volume in the prefrontal, OFC and temporal cortices, hippocampus and parahippocampal gyrus, and lower WM volumes in the brain stem, cerebellum, medial temporal and frontal cortex. - mUPD vs. HC: showed more extensive lower GM and WM volumes in the OFC and limbic cortices. - DEL vs. mUPD: showed lower GM volume in the prefrontal and temporal cortices, and lower WM in the parietal cortex. - The first neurostructural imaging study of genetic subtypes about PWS.
3	Lukoshe et al. [64]	The Netherlands GE 3.0 T	Cortical reconstruction and volumetric segmentation	31/20 11 DEL 9 mUPD	12.3(DEL) 10.6(mUPD)	- Both DEL & mUPD: showed a decrease in brainstem volume and a tendency for cortical surface area and WM volume to become smaller, which are signs of impaired brain growth. The changes are inconsistent between subtypes. - mUPD: showed enlarged lateral ventricles, larger cortical CSF volume. as well as a trend towards increased cortical thickness; revealed reduced WM volumes in left superior and bilateral inferior frontal gyri, right cingulate cortex, and bilateral precuneus areas associated with DMN; showed signs of early brain atrophy. - DEL: showed small cerebellum, cortical and subcortical GM volumes.
4	Lukoshe et al. [65]	The Netherlands GE 3.0 T	SBM	35/24 12 DEL 12 mUPD	12.6 (DEL) 11.5 (mUPD)	- PWS vs. HC: showed lower lGI in four large clusters. comprising frontal, parietal and temporal lobes; showed lower cortical surface area in clusters with lower lGI; lGI correlated significantly with cortical surface area but not with cortical thickness; lGI in both hemispheres correlated with total IQ and verbal IQ; underlies cognitive impairment and developmental delay. - mUPD vs. DEL: had two small clusters with lower lGI in the right hemisphere; lGI of these clusters were correlated with cortical surface area.
5	Xu et al. [55]	China Siemens 3.0 T	Structure covariant	48/12 12 PWS 18 OB 18 HC	7.2 (PWS) 9.0 (OB)	- Both PWS & OB vs. HC: showed 10 co-varying brain regions in bilateral dorsolateral and medial prefrontal cortices, right anterior cigulate cortex and bilateral temporal lobe; reduced FA of GM fibers connected to the 10. - PWS vs. OB: showed decreased GM volume, especially in hypothalamus.
6	Manning et al. [66]	UK --	VBM/ GLM	60/20	23.10	- PWS vs. HC: showed large and widespread bilateral clusters of both increased and decreased GM volume; increased cortical thickness results in increased cortical volume; analysis of myelin content using magnetization transfer saturation was broadly similar, with the exception of highly localized areas (insula). - Discussed possible developmental and maturational mechanisms of PWS.
7	Ge et al. [67]	China --	VBM	102 75 DEL 27 mUPD	newborns	- mUPD vs. DEL: the most variable GM volumes between the two genotypes are MFC, OFC and inferior frontal gyrus; explained the differences in maladaptive behaviors and emotions.
8	Yamada et al. [68]	Japan GE 3.0 T	——	61/21	21.0	- PWS vs. HC: significantly reduced TIV; decreased relative lobular volume ratios in posterior inferior lobules and increased bilateral dentate nuclei ratios with age, sex, and TIV as covariates; altered lobular volume ratios were negatively correlated with overeating and autism and positively correlated with obsessive and intellectual symptoms. - The first neurostructural imaging study of volume differences in cerebellar structures of PWS.
9	Caixàs et al. [69]	Spain 1.5 T	VBM	162/30 30 PWS 132 ID	27.5	- PWS vs. ID: the imitation of motor actions is significant impairment and constructional praxis domain is relatively preserved; increased GM volume in sensorimotor and subcortical regions.

**Table 3 jcm-12-01054-t003:** Studies of dMRI in PWS.

Number	Author	Country and Equipment	MRI Assessment Method	Subjects (*n*) Total/PWS	PWS Mean Age (Years)	Key Findings
1	Yamada et al. [77]	Japan GE 3.0 T	DTI	16/8	19	- PWS vs. HC: trace value was significantly higher in the left frontal WM and the left dorsomedial thalamus; FA was significantly reduced in the bilateral PLIC, the right frontal WM, and the splenium of the corpus callosum. - The first DTI study of PWS.
2	Rice et al. [78]	Australia GE 3.0 T	DTI TBSS	30/15	21	- PWS vs. HC: decreased FA value in the splenium of the corpus callosum and internal capsule; these WM lesions are related to orientating attention, emotion recognition, semantic processing, and sensorimotor dysfunction.
3	Xu et al. [55]	China Siemens 3.0 T	DTI 24 directions	48/12 12 PWS 18 OB 18 HC	7.2 (PWS) 9.0 (OB)	- Both PWS & OB vs. HC: showed 10 co-varying brain regions in bilateral dorsolateral and medial prefrontal cortices, right anterior cigulate cortex and bilateral temporal lobe; reduced FA of GM fibers connected to the 10.
4	Lukoshe et al. [79]	The Netherlands GE 3.0 T	DTI TBSS	89/28 15 DEL 13 mUPD	14.4 (DEL) 11.8 (mUPD)	- mUPD vs. both DEL & HC: decreased FA value in the corpus callosum, cingulate and superior longitudinal fasciculus. - mUPD are similar to the WM abnormalities in individuals with psychotic disorders, associated with primary myelin injury. - DEL are consistent with their substantially lower risk of psychosis.

**Table 4 jcm-12-01054-t004:** Studies of t-fMRI in PWS.

Number	Author	Country and Equipment	MRI Measures and Analysis	Subjects (*n*) Total/PWS	PWS Mean Age (Years)	Key Findings
1	Shapira et al. [81]	USA GE 3.0 T	TCA; glucose ingestion overnight	3 PWS	36.7	- PWS vs. OB: showed a significant delay in activation at the hypothalamus and other brain regions associated with satiety (i.e., the insula, VMPFC and nucleus accumbens).
2	Holsen et al. [82]	USA Siemens 3.0 T	Visual food stimuli	18/9 5 PWS with pre-meal 4 PWS with pos-meal	14.7	- HC with pre-meal vs. HC with post-meal: greater activation to food pictures in AMY, OFC, MPFC and frontal operculum. - PWS with post-meal vs. PWS with pre-meal: greater activation to food pictures in OFC, MPFC, insula, hippocampus, and parahippocampal gyrus. - PWS vs. HC: greater activation in food motivation networks with post-meal.
3	Miller et al. [83]		Ingestion of an oral glucose load; Viewing pictures	16/8	25	- PWS vs. HC: showed increased activation in VMPFC when viewing pictures. - Supported the importance of neural pathways that guide reward related behavioral regulation of food responses in PWS.
4	Dimitropoulos et al. [84]	USA Siemens 3.0 T	Visual food stimuli	19/9	21.6	- PWS vs. HC: showed increased activation in neural circuitry regulating hunger and motivation (hypothalamus, OFC) in response to high- vs. low- calorie foods.
5	Holsen et al. [85]	USA Siemens 3.0 T	Visual food stimuli	18 PWS 9 mUPD 9 DEL II	24.4 (DEL) 20.3 (mUPD)	- PWS vs. HC: showed greater activity in response to food pre- and post-meal. - DEL vs. mUPD: increased food motivation network activation both pre- and post-meal, especially in mPFC and AMY; decreased activation in the DLPFC and parahippocampal gyrus post-meal. - DEL showed reduced behavioral inhibition around food, whereas UPD were more able to maintain cognitive control over food intake impulses.
6	Holsen et al. [86]	USA Siemens 3.0 T	Visual food stimuli	43/14 14 PWS 14 OB 15 HC	24.3	- PWS vs. Both OB &HC: higher activity in reward/limbic regions (NAc, amygdala) and lower activity in the hypothalamus and hippocampus in response to food (vs. non-food) images pre-meal; higher subcortical activation (hypothalamus, amygdala, hippocampus) post-meal.
7	Klabunde et al. [87]	USA GE 3.0 T	Record skin picking episodes	17 PWS 10 with episodes	15.7	- Skin picking vs. non-skin picking: higher activity regions involved in interoceptive and somatosensory processing. Skin picking severity was significantly negatively correlated with mean activation within the right insula and left precentral gyrus.

**Table 5 jcm-12-01054-t005:** Studies of rs-fMRI in PWS.

Number	Author	Country and Equipment	MRI Measures and Analysis	Subjects (*n*) Total/PWS	PWS Mean Age (Years)	Key Findings
1	Zhang et al. [88]	China Siemens 3.0 T	ALFF	39/21	7.3	- PWS vs. HC: ALFF: selected 10 ROIs related four RSNs: DMN, core network, prefrontal network and motor sensory network. - FC: between 10 ROIs: further study obesity model of PWS.
2	Zhang et al. [89]	China Siemens 3.0 T	ALFF GCA	39/21	9.3	- PWS vs. HC: increased ALFF value in ACC, left AMY and hypothalamus; decreased ALFF value in MPFC and right AMY; increased GCA in the directions from both the left AMY and right AMY to hypothalamus, from ACC to right AMY, from MPFC to both left AMY and right AMY, and from ACC to MPFC. - To investigate the causality of key neural pathways involved in overeating in PWS.
3	Pujol et al. [90]	Spain Siemens 1.5 T	seed based	53/24	26.3	- Based on 10 ROIs involving in the dorsal and ventral aspects of the caudate nucleus and putamen to explain OCD, self-picking and obsessive eating behavior. - PWS vs. HC: abnormal FC between the prefrontal cortex and basal ganglia was correlated with the severity of OC behaviors; abnormal FC between the primary sensorimotor cortex and putamen loop was strongly associated with self-picking.
4	Tauber et al. [91]	France --	ICA	18 PWS with OXT 18 PWS without OXT	newborns	- PWS with OXT vs. PWS without OXT: significant improvements in Clinical Global Impression scale scores, social withdrawal behavior and mother–infant interactions; alterations of the right superior OFC network were correlated with changes in sucking and behavior.

ACC, Anterior cingulate cortex; ALFF, amplitude of low frequency fluctuations; AMY, amygdala; brain-PAD: Brain-predicted age difference; CSF, cerebrospinal fluid; DLPFC, dorsolateral prefrontal cortex DMN, default mode network; dMRI, diffusion magnetic resonance imaging; DTI, Diffusion tensor imaging; FA, fractional anisotropy; FC, functional connectivity analysis; GCA, Granger causality analysis; GE, General Electric Company; GLM, general linear model; GM, gray matter; HC, healthy controls; ICA, independent component analysis; ID, intellectual disability; IQ, intelligence quotient; lGI, Local gyrification index; MFC, the middle frontal cortex; mMRI, morphological magnetic resonance imaging; MPFC, medial prefrontal cortex; OB, obese subjects; OFC, the orbitofrontal cortex; OXT, oxytocin treatment; PLIC, posterior limb of the internal capsule; PWS, Prader–Willi syndrome; ROI, region of interest; rs-fMRI, rest-state functional magnetic resonance imaging; TBSS, tract-based spatial statistic; TCA, Temporal clustering analysis; t-fMRI, task based functional magnetic resonance imaging; TIV, total intracerebellar volume; VBM, the voxel-based morphometry; VMPFC, ventromedial prefrontal cortex; WM, white matter.

## Data Availability

Not applicable.

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
