# Peer review of "Progress in Brain Magnetic Resonance Imaging of Individuals with Prader–Willi Syndrome"

_jcm, 2023, doi:10.3390/jcm12031054_

Round 1
Reviewer 1 Report
The authors present a nicely written review of MRI in Prader-Willi-Syndrome. The manuscript is well organized and different MRI modalities were described in subsections separately.
However, there is just one aspect missing. What is the benefit of MRI in PWS? Of course, using MRI altered brain regions and mechanisms will be detected, but why could this be important?
I would suggest to add a "therapy" section and figure out what MRI results may contribute to improve everyday life of individuals with PWS.
Author Response
Response to Reviewer 1 Comments
Point 1: What is the benefit of MRI in PWS? Of course, using MRI altered brain regions and mechanisms will be detected, but why could this be important?
Response 1: Thank you for your advice. We have made adjustments accordingly. We agree the benefit of MRI in PWS is important and necessary, and we have added it according. On the one hand, MRI technique had the advantages of being non-invasive, free of ionizing radiation, as well as multiparametric and multimodal. On the other hand, brain MRI in PWS provide useful structural and functional information on the underlying neuropathological process of PWS-related cognitive dysfunctions and provide supporting evidence for further molecular biology studies.
Line 47-49:
In section 1 “Introduction”:
“Recently, multimodal MRI studies in PWS have provided useful brain structural and functional informations on the underlying pathophysiological mechanisms, allowing for the development of objective and reliable neuroimaging markers.”
Line 128-129:
In section 5 “MRI”:
“These findings provide an important role for further treatment, care and prognosis.’’
Line 164-169:
In section 5.2 “PWS-related MRI methods”:
“Neuroimaging methods have been used to reveal brain abnormalities in PWS, including positron emission tomography (PET), magnetic resonance imaging (MRI) and magnetoencephalography. Among them, brain MRI in patients with PWS has the advantages of being radiation-free, providing multiparameter options, and having high tempo-spatial resolution. ”
Point 2: I would suggest to add a "therapy" section and figure out what MRI results may contribute to improve everyday life of individuals with PWS.
Response 2: Thank you for your reminder, we have made adjustments accordingly. After reviewing the past literature, we found that MRI studies mainly play a diagnostic role for PWS, which in turn guides the later treatment and prognosis. There was only one article that studied MRI changes in PWS before and after drug treatment (Tauber et al., 2017), and we describe it in more detail in our manuscript accordingly.
Line 413-416:
In section 5.2.4.2 “Other behaviors”:
“Tauber et al. [94] found increased connectivity of the right superior orbitofrontal network after oxytocin treatment in infants with PWS based on independent component analysis (ICA), which was correlated with improved oral motor skills and social behaviors. This was the only MRI study that focused on treatment.”
Line 444-449:
In section 6 “Challenges and future prospects”:
“Third, recent research in PWS has focused more on intake and OC behavior and less on longitudinal studies in younger age groups, pre- and post-treatment comparisons of growth hormone, and other cognitive impairment and comorbidity studies. Among them, brain alterations before and after PWS drug treatment should be focused on to facilitate future prognosis and management. ”

Reviewer 2 Report
The manuscript ‘Progress in Brain Magnetic Resonance Imaging of Individuals with Prader-Willi Syndrome’ presents a literature review of papers on the use of standard MRI sequences to characterize the Prader-Willi Syndrome (PWS). The manuscript is of interest and well written. It is acceptable in its current format, although I would like to make few minor suggestions on how it could be improved.
1- In my opinion, lines 46- 63 (including Figure 1) are out of place, since they refer to exploration methods instead to the disease. Also, the most of 5.1 are a description of PWS phenotyping, not directly related to MRI sequences. I would suggest to make a restructuration of the information, making a longer introduction split in two parts a) about pathology and b) about imaging (MRI) techniques.
2- In the framework description (points 1-4), the authors made an overview about PWS that I found, as neophyte in that pathology, very interesting. However, it may be too detailed for more expert readers. Depending on the target public, it might be summarized.
3- Tables are difficult to read I would suggest to include horizontal lines between items or bullet points to make it easier to read
Author Response
Response to Reviewer 2 Comments
Point 1: In my opinion, lines 46- 63 (including Figure 1) are out of place, since they refer to exploration methods instead to the disease. Also, the most of 5.1 are a description of PWS phenotyping, not directly related to MRI sequences. I would suggest to make a restructuration of the information, making a longer introduction split in two parts a) about pathology and b) about imaging (MRI) techniques.
Response 1: Thanks for your suggestions.
- Wetotally agree that Figure 1 is more suitable for MRI methods part, so we adjusted this figure and its description to section 2 “PWS-related MRI methods” to help describe the MRI sequence in detail.
Line 169-176:
In section 5.2 “PWS-related MRI methods”:
“The MRI methodological flow chart of PWS is shown in Figure 1.”
Figure 1. Schematic figures of brain magnetic resonance imaging (MRI) analysis of individuals with Prader-Willi syndrome (PWS). (A) The PWS and control groups are included and MRI scans are performed. (B) The raw MRI acquisition images are obtained. (C) Data processing are performed on the acquired MR images. (D) Statistical difference in brain structure and function are obtained by comparison between groups.
- We totally agree to make a restructuration of section 5.1 “PWS-related brain regions”, which is important and a bridge between MRI and PWS phenotypes. PWS-related brain regions are generally delineated by neuroimaging-based atlas.MRI is the most classical technique for neuroimaging, so this part is suitable for this section, but it needs to be restructured as your suggestion. Brain regions also influenced by pathology of PWS, we have discussed the pathology and physiological mechanisms of PWS in section 3 “Pathophysiological mechanisms”.
Line 128-131:
In section 5 “PWS”:
“Hence, the PWS-related brain regions and the PWS-related MRI methodology are both important for understanding the pathophysiological mechanisms of PWS-related phenotypes.”
Line 133:
In section 5.1 “PWS-related brain regions”:
“PWS-related brain regions are generally delineated by neuroimaging-based atlas.”
Line 132: section 5.1 “PWS-related brain regions”
Line 164: section 5.2 “PWS-related MRI methods”
Line 177: section 5.2.1 “ Conventional structural MRI”
Line 194: section 5.2.2 “ Morphological MRI (mMRI)”
Line 283: section 5.2.3 “ Diffusion MRI (dMRI)”
Line 329: section 5.2.4 “ Functional MRI (fMRI)”
Line 357: section 5.2.4.1 “ Overeating”
Line 357: section 5.2.4.1 “ Other behaviors”
Point 2: In the framework description (points 1-4), the authors made an overview about PWS that I found, as neophyte in that pathology, very interesting. However, it may be too detailed for more expert readers. Depending on the target public, it might be summarized.
Response 2: Thank you for your advice. We have summarized more concisely and clearly the descriptions in points 1-4 accordingly, especially the specialized descriptions in the section 3 “pathophysiological mechanisms” .
Line 57-65:
In section 2 “Molecular genetics”:
“(1) DEL: a microdeletion of the paternal genes at the chromosomal 15q11-q13 locus (approximately 65-70%), further divided into Type â… and Type â… â… ; (2) mUPD: subsequent to trisomic rescue (approximately 20-30%) in meiosis; (3) imprinting center (IC) mutations in DNA fragments (4%); and (4) balanced translocation of chromosome 15 (<1%) [12-14].”
“ Compared to patients with DEL subtype of PWS, patients with other subtypes of PWS are more prone to have psychotic episodes and anxiety symptoms, have higher verbal intelligence quotient scores and have better social reasoning skills [15-18].”
Line 76-89:
In section 3 “Pathophysiological mechanisms”:
“The two components of hypothalamus have both orexigenic and anorexigenic neuronal populations, participating in the regulation of feeding behaviors [23-25]. In addition, hypothalamic-pituitary axis-related peptide hormones-expressing neurons involve in energy metabolism [23,24,26,27]. That is, dysfunction of hypothalamic neuropeptides and hormones results in the metabolic phenotype of PWS, including extreme overeating, obesity, and other clinical features [28-31]. Contrary to the above study results, another study found no abnormalities in related neurons of the hypothalamus in patients with PWS [32]. Of note, another theory about excitation-inhibition imbalance mechanism focusing on classical transmitters (i.e., amino acids, acetylcholine and monoamines). Classical transmitters are not closely linked to the hypothalamus and are used to understand clinical phenotypes besides obesity, such as OC behaviors, psychiatric problems and some autonomic deficits, in patients with PWS [33,34]. Gamma-aminobutyric acid (GABA) is an amino acid. Genes encoding of GABA type-A (GABAA) receptor are localized on chromosome 15 [35].”
Line 100-102:
In section 4 “Clinical manifestations”:
Classical nutritional stages in PWS are poor feeding in early neonatal life and consequent overwhelming appetite leading to morbid obesity [39-41] (Table 1).
Line 105-108:
In section 4 “Clinical manifestations”:
“Hyperphagia in PWS manifests as an extreme desire to eat, a lack of normal satiety and confusion about food choices [44-46]. A prior study suggested that excessive preoccupation with food might be an overexpression of the joviality of eating [47].”
Line 105-108:
In section 4 “Clinical manifestations”:
“Children with PWS may take twice as long to achieve physical and social milestones compared to healthy individuals, then progress to having mild intellectual disability [49].”
Line 105-108:
In section 4 “Clinical manifestations”:
“Obsessive compulsive (OC) behaviors in patients with PWS, such as skin picking, bring pleasure and comfort, which are slightly different from the typical tendencies of OC disorder [50].”
Point 3: Tables are difficult to read I would suggest to include horizontal lines between items or bullet points to make it easier to read
Response 3: Thank you for your careful review and constructive suggestions. We have added short horizontal lines in front of each item in the findings in tables to make it clearer to read, and the short horizontal lines are marked in red font in the table.
